# Active Touch Sensing for Robust Hole Detection in Assembly Tasks

**DOI:** 10.3390/s25154567

**Published:** 2025-07-23

**Authors:** Bojan Nemec, Mihael Simonič, Aleš Ude

**Affiliations:** Humanoid and Cognitive Robotics Lab, Jožef Stefan Institute, 1000 Ljubljana, Slovenia; mihael.simonic@ijs.si (M.S.); ales.ude@ijs.si (A.U.)

**Keywords:** active tactile sensing, Peg-in-hole assembly, 3D object localization and map registration, deterministic and probabilistic search algorithms, robustness to sensor and map uncertainty

## Abstract

In this paper, we propose an active touch sensing algorithm designed for robust hole localization in 3D objects, specifically aimed at assembly tasks such as peg-in-hole operations. Unlike general object detection algorithms, our solution is tailored for precise localization of features like hole openings using sparse tactile feedback. The method builds on a prior 3D map of the object and employs a series of iterative search algorithms to refine localization by aligning tactile sensing data with the object’s shape. It is specifically designed for objects composed of multiple parallel surfaces located at distinct heights; a common characteristic in many assembly tasks. In addition to the deterministic approach, we introduce a probabilistic version of the algorithm, which effectively compensates for sensor noise and inaccuracies in the 3D map. This probabilistic framework significantly improves the algorithm’s resilience in real-world environments, ensuring reliable performance even under imperfect conditions. We validate the method’s effectiveness for several assembly tasks, such as inserting a plug into a socket, demonstrating its speed and accuracy. The proposed algorithm outperforms traditional search strategies, offering a robust solution for assembly operations in industrial and domestic applications with limited sensory input.

## 1. Introduction

In robotics, various sensors are employed to enable the execution of complex tasks [1]. These sensors include 2D and 3D cameras, force sensors, tactile sensors, laser scanners, and similar devices [2]. Among these, cameras have proven to be particularly effective and cost-efficient for applications such as bin picking [3], automated robot assembly [4], and quality control [5]. However, challenges arise when objects are not visible due to overlap, poor lighting conditions, or suboptimal camera positioning [6,7]. Accurate hand–eye camera calibration is a critical yet frequently underestimated challenge in robotics, which can present problems for operations requiring high precision, such as when assembling objects with low tolerance [8].

In such cases, reliance must shift to alternative sensors, such as force and touch sensors, which do not provide information as rich or comprehensive as that from cameras [9,10]. This paper addresses this issue and proposes a tactile localization method that does not rely on cameras.

A typical assembly operation, such as inserting a peg into a hole, can be divided into two primary phases: positioning the peg near the opening and the actual insertion. This paper does not address the insertion phase, which requires force sensors and that has already been well studied in robotics [11,12,13,14]. Instead, we focus on the approach phase, i.e., the localization of the opening, using only force sensors or touch detection.

Several heuristic and statistical search methods are commonly employed for this purpose, including random search [15], spiral search [16], genetic algorithm-based search [17], ergodic search [18], and others. Among them, only random and spiral searches do not require prior knowledge of the environment. In contrast, ergodic search utilizes a probability distribution indicating where the assembly object is likely to be located in space, making it more effective than the other two methods.

Rather than relying primarily on heuristics or statistical priors, our approach assumes the availability of an accurate geometric model of the environment, although its position and orientation relative to the robot are initially unknown. The goal is to localize this model through systematic exploration.

## 2. The State of the Art

The problem of object localization in assembly operations has been widely studied in prior research, with diverse approaches proposed depending on available sensing modalities and application contexts. Early methods, such as those by [19,20], utilized pre-acquired contact maps combined with particle filters to enable precise localization using sparse tactile data. Similarly, ref. [21] introduced a computationally efficient iterative Bayesian Monte Carlo technique for six degree-of-freedom (6-DOF) pose estimation, demonstrating robustness in tactile localization tasks. Other approaches, such as the Gaussian mixture model-based contact state detection method proposed by [22], leverage wrench signals to facilitate peg-in-hole assembly localization.

Building on these foundations, tactile sensing for object localization has been further advanced by [23], who introduced the Next Best Touch (NBT) strategy to identify the most informative subsequent contact for efficient pose estimation. Extensions of this concept to 2D visual maps were explored by [24] using recursive Bayesian filtering to estimate belief distributions over possible locations, with [25] refining this framework to address both localization and shape uncertainty in active tactile sensing. Recent works have incorporated deep learning techniques to process tactile data more effectively; for example, refs. [26,27] demonstrated the use of deep neural networks (DNNs) for tactile object pose estimation from high-resolution sensor arrays, achieving significant accuracy improvements. Other studies, such as [9,28], have successfully applied tactile contact sensing for object recognition and classification, highlighting the growing capabilities of tactile perception.

In parallel, related research in robotic grasping and manipulation has emphasized the integration of multimodal sensory inputs, combining vision, force, and tactile data to enhance pose estimation accuracy and robustness under uncertainty [29,30].

Despite these significant advancements, the majority of existing work—apart from [19,20]—does not explicitly target the challenge of assembly pose search using sparse binary touch sensors, which provide extremely limited and discrete information. This sparse sensing modality imposes unique challenges in developing algorithms capable of robust, efficient localization under minimal sensory input. Consequently, this remains a critical open problem in automated assembly, motivating further research into probabilistic and adaptive methods tailored for sparse tactile feedback.

Binary touch sensing, despite its simplicity, offers several key advantages in constrained environments. Unlike visual-tactile sensing, which requires cameras with clear line-of-sight, adequate lighting, and often precise calibration between visual and robot coordinate frames, binary contact sensors can operate in complete darkness, through occlusions, and without complex setup. This makes them particularly well-suited for tasks where cameras cannot be reliably deployed, such as operations in enclosed fixtures, poorly illuminated areas, or behind physical obstructions. Furthermore, visual-tactile systems generally require high-fidelity calibration and often depend on higher-bandwidth communication and processing pipelines, whereas binary touch sensing enables lightweight, reactive implementations that are easier to deploy and maintain in industrial environments. These trade-offs motivate the development of efficient localization algorithms that rely solely on binary tactile feedback.

## 3. Materials and Methods

In this section, we present our original algorithms for detecting the 3D position of objects using touch sensing. We begin by introducing a basic search algorithm for 2D position detection and subsequently extend it to handle 3D position estimation. We then enhance these algorithms with a probabilistic search framework designed to robustly manage sensor noise, inaccuracies in the object map, and variations due to object rotation.

### 3.1. Map Registration

In this section, we present a deterministic 2D search method that serves as a foundation for 3D search, introduced in Section 3.2, and its further enhancement into a probabilistic framework, described in Section 3.3. Our approach shares similarities with Next Best Touch (NBT) methods [23], as it systematically refines the search region through geometric region elimination in consecutive steps.

To help orient the reader, we briefly describe the intuition behind the proposed method before delving into the algorithmic details. The robot, equipped with a touch sensor, can detect when it comes into contact with a surface and can measure the height of the contact point. From this information, it knows which predefined surface region of the object was touched (as each has a distinct height), but not the exact x-y location within that region. By analyzing the relative distances and directions between consecutive touch points, and aligning these with a known 3D model of the object, the algorithm incrementally narrows down the possible regions where the contact could have occurred. This process continues iteratively, pruning inconsistent hypotheses and refining the estimated position. Once the robot has localized one of the contact points with sufficient confidence, it can infer the relative position of the goal (e.g., the center of a hole) and successfully complete the insertion.

We assume the availability of a 3D map of the object where the assembly operation takes place. Furthermore, we consider that the 3D object consists of a finite number of horizontal faces (quasi-iso-height regions). These surfaces are represented as a 2D model map $M={\left\{S_{i}\right\}}_{i=1}^{N}$ in the *x*-*y* plane, where $S_{i}$ denotes the partitions of the map and *N* is the number of partitions. Each partition is defined as an area of the object having the same height $z_{i}^{m}$ when put on the horizontal surface, $S_{i}={\left\{p_{i,j}^{m}\right\}}_{j=1}^{N_{i}},p_{i,j}^{m}={\left[x_{i,j}^{m},y_{i,j}^{m},z_{i}^{m}\right]}^{T}$, and $N_{i}$ is the number of discrete points within the partition $S_{i}$. In practice, we obtain these points by discretization directly from a CAD model or, alternatively, using a scanner device. An example of such a region-based map is depicted in Figure 1.

While the 2D map M representing the object’s geometry and the object’s orientation is known, the position of the object in the robot’s coordinate system is unknown. There are many practical examples in industry that satisfy these requirements, for example, all objects that are rotationally invariant. There is also a common case when we can provide the exact orientation of the object, but not its position. We consider a scenario where the robot must determine any point on the target region $S_{g}$, with the centroid denoted by $p_{g}^{m}$ in the map coordinate system. Initially, we are given an estimate of a point above the target region $S_{g}$ in the robot’s coordinate system, which we denote here by ${\tilde{p}}^{r}\left(0\right)$. However, due to uncertainty in this initial position, the robot might initially contact a different region. Note that we are not specifically looking for the centroid of $S_{g}$ but for any point in $S_{g}$.

Next, the robot moves in the −z direction of the map coordinate system until it makes contact with the object surface. By computing the *z*-coordinate of this initial contact point $p^{r}\left(0\right)$ in the map coordinate system (see Equation (Equation 4)), the robot determines which region has been touched. (Due to inaccuracies in robot positioning, the sensed *z*-coordinate, denoted $z_{s}$, may not exactly coincide with any of the predefined region heights $z_{1},z_{2},\ldots ,z_{N}$ in the map. In such cases, the contact is assigned to the region whose nominal height $z_{i}$ minimizes $|z_{s}-z_{i}|$. This assignment procedure assumes the existence of a known transformation between the robot’s coordinate system and that of the map, so that $z_{s}$ and $z_{i}$ can be meaningfully compared. This assumption is made here to simplify the explanation, but it is removed in later sections by learning the vertical offset of the robot frame.) We denote the touched region as $S_{s}\left(0\right)=S_{i}$, where *i* is the index determined by the measured height. The region Si is defined as the set $p_{j}^{m}{j=1}^{N_{i}}$, with each point $p_{j}^{m}\in S_{i}$ also belonging to $S_{s}\left(0\right)$.

In the following, we apply notation where vectors with superscript ${\left(.\right)}^{r}$ are expressed in the robot’s coordinate system, while the corresponding vectors with the superscript ${\left(.\right)}^{m}$ are expressed in the map’s coordinate system.

Initially, we determine the touched position in the map coordinate system as a point closest to the centroid of the touched region and ensure it is also contained in that region. We denote this position as $p_{e}^{m}$. The algorithm then computes the displacement vector:(1)$$d^{m}=p_{g}^{m}-p_{e}^{m}.$$
Next, the robot moves to the next estimate of the position above the target region:(2)$${\tilde{p}}^{r}\left(1\right)={\tilde{p}}^{r}\left(0\right)+R_{0}d^{m},$$
where the rotation matrix $R_{0}\in R^{3\times 3}$ accounts for the rotation between the robot and the map coordinate system. The robot then moves again along the −z-coordinate of the map coordinate system until it touches the surface of the object. The *z*-coordinate of the new contact point $p_{r}\left(1\right)$ in the map coordinate system determines the next touched region $S_{t}\left(1\right)$.

To refine the estimate, we update the search region $S_{s}\left(0\right)$ by selecting all points $p^{m}$ within $S_{s}$ that satisfy:(3)$$S_{s}\left(1\right)=\left\{p_{j}^{m}\in S_{s}\left(0\right)|p_{j}^{m}+d^{m}\in S_{t}\left(1\right)\right\}.$$
The updated search region $S_{s}\left(1\right)$ contains only the points that fulfill the above condition. The next estimate of $p_{e}^{m}$ is computed as the centroid of $S_{s}\left(1\right)$. Like before, if the centroid is not contained within $S_{s}\left(1\right)$, we take a random point from $S_{s}\left(1\right)$ as the estimate of $p_{e}^{m}$. This operation is repeated until the robot hits a point on the target region $S_{g}$. We denote the iteration index by *k*.

In Appendix B, we show that the last touched position $p^{r}\left(k\right)$ is guaranteed to lie within the target region $S_{g}$.

The above procedure defines an iterative algorithm outlined in Algorithm 1. The steps in Algorithm 1 correspond to the iterative narrowing process described earlier. At each iteration, the robot uses its latest touch input to update its hypothesis about the object’s position by eliminating physically inconsistent regions based on the known geometry of the object.
**Algorithm 1:** Map registration algorithm using touch sensing
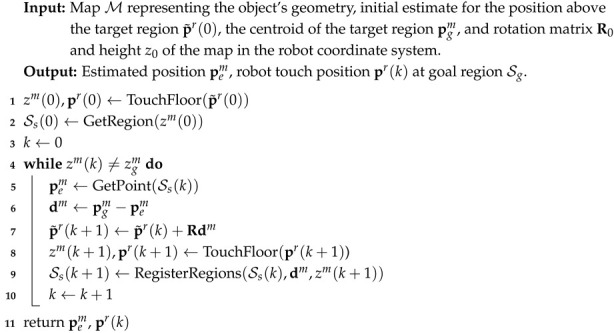


In Algorithm 1, we apply the following functions:TouchFloor is a function that involves the motion of the robot from the initial position ${\tilde{p}}^{r}\left(k\right)$ along the −z axes in the map coordinate system until it touches the surface of the object. It also computes the $z^{m}\left(k\right)$-coordinate of the touch point in the map coordinate system. This calculation involves the transformation(4)$$\left[\begin{array}{c} x^{m}\left(k\right)\\ y^{m}\left(k\right)\\ z^{m}\left(k\right) \end{array}\right]=R_{0}^{T}\left(p^{r}\left(k\right)-\left[\begin{array}{c} 0\\ 0\\ z_{0} \end{array}\right]\right),$$
where $z_{0}$ is a constant that defines the *z*-component of the map coordinate system origin expressed in robot coordinates. Note that *x*- and *y*-coordinates of the map coordinate system origin are unknown.GetRegion returns the region index based on the measured $z^{m}\left(k\right)$-coordinate at the contact point.GetPoint returns a point from $S_{s}\left(k\right)$ closest to the centroid of $S_{s}\left(k\right)$.RegisterRegions returns the region composed of all points $p_{j}^{m}$ that satisfy the condition $p_{j}^{m}\in S_{s}\left(k\right),\;\;p_{j}^{m}+d^{m}\in S_{t}\left(k+1\right)$.

To demonstrate the effectiveness of the proposed method, we apply it to the task of locating the socket into which a robot must insert an audio jack plug, as illustrated in Figure 2. The socket is a cylindrical structure with a radius of 10mm and a central hole of radius 2.5mm. The search area spans 40×40mm. The iterative refinement process is shown in Figure 3, where the object is initially offset from its ideal position by −4.8mm along the *x*-axis and −8.8mm along the *y*-axis. Consequently, the robot misses the hole in the first attempt, as indicated by the red circle in the first sub-figure at step k=0. Observe how the search area $S_{s}(k)$ (shown in white) is progressively reduced until the estimated position $p_{e}^{m}$ lies within the target region $S_{g}$, thereby enabling successful plug insertion.

### 3.2. Map Registration with Unknown Object Base Plane Height

The algorithm presented in the previous section assumes that the *z*-coordinate of the object’s surface can be directly determined from the touch sensor’s reading. In other words, it requires prior knowledge of the height of the object’s base plane in the robot coordinate system so that each touch immediately reveals which region was contacted. However, if the exact height is unknown, the robot cannot directly ascertain which region it has touched. In such scenarios, estimating the object’s base *z*-coordinate (height) becomes a necessary step before proceeding with precise localization. The algorithm presented in this section overcomes this limitation by eliminating the need for prior height information, thus ensuring that the robot can still identify the contacted region.

We propose an iterative algorithm to estimate an object’s base height using a 3D map and successive touch operations. As, before, we assume that the object consists of a finite number of uniform height regions, denoted as $S_{i}$, where i={1,…,N} denotes the region index, each located at a distinct height $z_{i}^{m}$. From the 3D map, the algorithm first identifies the number of these regions, *N*, and their corresponding heights $z_{i}^{r}$ in the robot coordinates.

The algorithm begins by selecting an arbitrary position above the object, establishes the contact point $p_{0}^{r}$ using the TouchFloor procedure and records the *z*-coordinate as height $z_{0}^{r}$. At this stage, it is unclear which of the map’s regions $S_{i},i=\left\{1,\ldots ,N\right\}$ the robot has touched. Therefore, the algorithm initializes a candidate region $S_{s,i}$ for each *i*, effectively treating all *N* regions as potential matches. In subsequent steps, the algorithm narrows down the feasible candidate regions by eliminating regions that are inconsistent with additional measurements.

The robot touches the object at another arbitrary point $p_{t}^{r}$, and a displacement vector in the map frame is computed as:(5)$$d^{m}=R\left(p_{t}^{r}-p_{0}^{r}\right).$$
The new contact point yields a height measurement $z_{t}^{r}$. We calculate height difference:(6)$$dz=z_{t}^{r}-z_{0}^{r},$$
For each candidate region, the height $z_{s,i}^{m}$ is updated as(7)$$z_{s,i}^{m}=z_{i}^{m}+dz,$$
where $z_{i}^{m}$ is the height (*z*-coordinate) of the *i*-th region $S_{i}$. Additionally, each candidate region $S_{s,i}\left(k\right)$ is updated by retaining only those points that satisfy the condition:(8)$$p^{m}\in S_{s,i}\left(k\right),\;p^{m}+d^{m}\in S_{h_{s,i}},$$
where $S_{z_{s,i}}\in M$ represents the set of regions at height $z_{s,i}$.

This process is repeated until all but one of the candidate region have been eliminated (i.e., their $S_{s,i}$ areas are reduced to zero). The remaining candidate region is then identified as the correct match for the initial contact point $p_{0}^{r}$ and the height associated with $S_{i}$ can be used to estimate the base *z*-coordinate of the object.

The algorithm is outlined in Algorithm 2. In addition to the functions already used in Algorithm 1, the following new functions are defined (in function TouchFloor, an unknown value $z_{0}^{r}$ appears; however, since the results of this function are subtracted in Algorithm 2, the value of $z_{0}^{r}$ does not affect the result and can be set to 0):Area($S_{i}$) returns the area of the region $S_{i}$.rand(m,n) returns a m×n matrix with random numbers.CountFeasibleRegions returns the number of feasible candidate regions, i.e., regions with area greater than 0.

The underlying intuition behind this approach is that once the robot touches all planes constituting the object, we can uniquely determine the identity of each plane. In practice, the identity of a certain plane can often be determined by touching only some of the planes. By tracking the sequence of detected planes and their relative displacements, the algorithm ensures reliable plane identification. This process is illustrated in Figure 4.

By estimating the *z*-coordinate before searching for the *x*- and *y*-coordinates (as described in Section 3.1), the algorithm significantly reduces the initial search space, minimizing computational complexity. Experimental results in Section 4 show that this additional step of determining the *z*-coordinate of the object’s base plane only marginally increases the total number of search iterations.
**Algorithm 2:** Map registation with unknown object base plane height using touch sensing
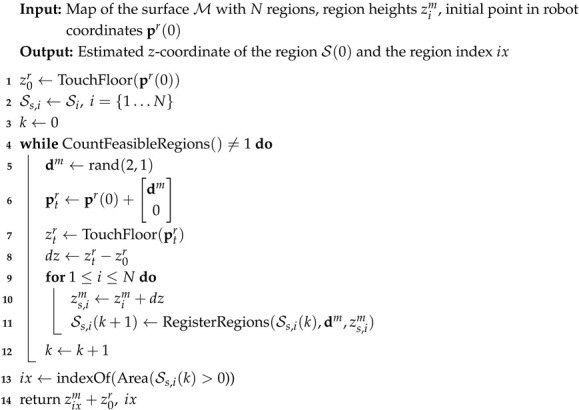


### 3.3. Probabilistic Map Registration

In real-world applications, robotic systems are often subject to various sources of uncertainty, including sensor noise, imperfect object maps, and calibration errors. While the deterministic version of the algorithm can tolerate moderate noise, it may fail when such deviations lead to the elimination of valid regions due to small inconsistencies. To address this, we introduce a probabilistic extension that models displacement as a distribution rather than a fixed value. Instead of rejecting inconsistent hypotheses outright, the probabilistic method assigns lower probabilities to less likely contact interpretations, allowing the algorithm to remain robust even when observations are noisy or partially inconsistent. This approach improves resilience without requiring major structural changes to the algorithm.

Unlike the deterministic approach, where the map M is partitioned in regions $S_{i}$, we now model the likelihood that a point belongs to the search region. Let $P(p^{m}\in S_{s}\left(k\right))$ denote the probability that a point $p^{m}$ belongs to the search region $S_{s}\left(k\right)$ at *k*-th iteration. Rather than this, the algorithm selects $p_{e}^{m}\in S_{s}\left(k\right)$ with the highest probability P.

In the deterministic map registration algorithm, the displacement vector $d^{m}$ is computed according to Equation (Equation 1). In the probabilistic framework, we instead model the displacement length ∥d∥ as a random variable with a continuous probability distribution. It is sampled from the range $\left[{∥d∥}_{min},{∥d∥}_{max}\right]$ assuming a normal distribution $N(\mu_{d},\sigma_{d})$, where $\mu_{d}$ is taken as the displacement length calculated with Equation (Equation 1) (see Figure 5). The parameter $\sigma_{d}$ models the uncertainty in the robot’s position and maps inaccuracies by controlling the spread of the Gaussian distribution used to sample the displacement length ∥d∥. Intuitively, $\sigma_{d}$ defines the width of this distribution, determining how broadly the search region is updated around the expected displacement. Typically, it is chosen such that the Gaussian covers approximately 20–30% of the nominal displacement vector length ∥d∥. At this scale, the Gaussian falls to about 5% of its peak height at the distribution’s edges, ensuring that the probabilistic update accounts for realistic positional errors without overly broadening the search space. This setting balances robustness against robot and map uncertainties with the efficiency of the search, and while the exact choice can be tuned experimentally, the described range provides a principled guideline.

The search region is updated accordingly. It is obtained by marginalizing over all possible displacement lengths. That is, instead of using a single displacement vector, we integrate the effect of sampled displacements weighted by their probability.

For computational reasons the length d on the interval $\left[{∥d∥}_{min},{∥d∥}_{max}\right]$ is divided into $N_{d}$ intervals, $d_{n},\;n=1\ldots N_{d}$, each with an associated probability $P_{d,n}$, providing that$$\sum_{n=1}^{N_{d}}P_{d,n}=1.$$
In each *k*-th search step, for each length $d_{n}$, we obtain region $S_{s,n}\left(k+1\right)$ using the Equation (Equation 3), following the same procedure as in the deterministic case. This way, we obtain $N_{d}$ regions $S_{s,n}\left(k+1\right)$ and compute:(9)$$S_{s}\left(k+1\right)=\overset{N_{d}}{\underset{n=1}{⋃}}S_{s,n}\left(k+1\right)$$
The probabilities are updated recursively as:(10)$$P\left(p^{m}\in S_{s}\left(k+1\right)\right)=P\left(p^{m}\in S_{s}\left(k\right)\right)\cdot \sum_{n=1}^{N_{d}}\left(P\left(p^{m}\in S_{s,n}\left(k+1\right)\right)\right),$$
Similar to before, in the deterministic approach, the algorithm narrows the search region $S_{s}\left(k\right)$ until the robot hits the goal region $S_{g}$.

Figure 6 illustrates the probabilistic map registration process for the audio pin insertion task, using of the same dimensions as in the deterministic method described in Section 3.1. In this scenario, the object is displaced from its ideal position by 4.6mm along the *x*-axis and 8.3mm along the *y*-axis. As a result, the robot misses the socket in the initial attempt at step k=0, and subsequently refines its estimate of the hole’s location over the following iterations.

## 4. Experimental Results

In this section, we experimentally validate the performance of the proposed algorithm and compare it with a random search strategy. For all experiments, we used a 7-DOF Franka Research 3 robot controlled by an enhanced Cartesian impedance control law. The applied control law is detailed in [31,32]. Enhancements to the original control law include bidirectional friction compensation, which improved positional accuracy for small displacements with low stiffness. The touch motion was implemented by setting the velocity command in the direction of the surface normal of the object and monitoring the force in the same direction. Motion was halted whenever the force exceeded a predefined threshold, and impact forces were mitigated by setting low stiffness in the impedance control law in the direction of the surface normal. A touching probe with a known geometry is attached to the tip of the robot. This allows us to determine the height of the touched point in the robot base coordinate system.

### 4.1. Inserting the Pin into the Socket

To validate the efficiency and robustness of our algorithm, we first replicated the experiment of inserting an audio pin into a socket, as described in Section 3.1. The experimental setup is depicted in Figure 7, where the socket was positioned on a table with its normal aligned along the *z*-axis. The socket was installed within a housing of 2 cm in diameter, with a socket hole measuring 3.5 mm.

The search area was confined to a 4×4 cm square, and the map M was encoded as a 400×400 matrix. Therefore, each point in the map corresponds to 0.1 mm. Since the coordinate frames of the map and the robot were aligned, the rotation matrix R was set to the identity matrix. The robot’s initial search position in robot coordinates was randomly selected within the defined search area.

In a set of 100 experimental trials, the algorithm successfully located the socket opening within one to ten attempts. Figure 8 illustrates the convergence behavior and standard deviation of the search process. In this experiment, the results are virtually identical when using deterministic or probabilistic search.

To further evaluate the algorithm’s performance under more challenging conditions, we considered a scenario where the object’s height relative to the robot is unknown. In this case, the height measurement alone is not sufficient to identify which of the map partitions has been touched. Therefore, the algorithm first estimates the correct *z*-position before proceeding with the *x*- and *y*-coordinate search, following the procedures outlined in Section 3.1 and Section 3.2, respectively. The convergence characteristics and standard deviation of this extended search process are illustrated in Figure 9.

A comparative analysis between Figure 8 and Figure 9 reveals that incorporating the additional *z*-coordinate search increases the maximum number of attempts by only two, while the average number of attempts increases marginally. This demonstrates that the added dimensional complexity does not significantly degrade the efficiency of the search algorithm.

As a benchmark, we conducted an additional 100 trials using a purely random search strategy within a 4×4 cm grid with a 0.2 mm resolution. To ensure fairness, no points were tested more than once. Figure 10 presents the convergence and standard deviation of the random search.

The results clearly highlight the superiority of our proposed search algorithm compared to random search. The algorithm demonstrates an average convergence speed more than six times faster than random search and exhibits significantly lower variance. In worst-case scenarios, our approach achieves over twenty times faster convergence, further validating its efficiency and reliability for real-world robotic assembly tasks.

In our final experiment, we evaluated the performance advantages of the probabilistic search algorithm under conditions of imprecise object mapping and positional inaccuracies of the robot. To simulate these uncertainties, we increased the distance parameter $d_{m}$ in Equation (Equation 2) by a factor of 1.2 while retaining the original value of $d_{m}$ in the registration process described by Equation (Equation 3). We then conducted 100 experimental trials of inserting an audio pin into a socket, comparing the success rates and convergence behavior of the deterministic and probabilistic search algorithms. The deterministic algorithm successfully inserted the pin into the socket in 85 out of 100 attempts, whereas the probabilistic algorithm achieved a success rate of 100 out of 100. Parameters $N_{d}$ and $\sigma_{d}$ were set to 20. These results, presented in Figure 11, clearly demonstrate the superiority of the probabilistic search algorithm in noisy environments, highlighting its robustness in handling uncertainties.

### 4.2. Inserting the Task Board Probe into the Socket

The subsequent experiment pertains to the *Task Board*, an internet-connected device designed to assess real-world robot manipulation skills [33]. Following the trial protocol, one of the operations involves extracting a probe from its socket, measuring the probe’s voltage level, wrapping the cable, and then stowing the probe. The last operation often fails due to factors such as incomplete grasp of the probe, significant movements during manipulation, environmental contact with the probe, and the effects of pulling the probe cable. As part of the euRobin project (https://www.eurobin-project.eu/, accessed on 20 July 2025), numerous Task Board manipulation solutions employing both in-hand and overhead cameras were introduced. However, these camera placements are inadequate for monitoring the probe-stowing operation. Consequently, an alternative solution utilizing touch detection was implemented for this purpose.

Initially, a 400 × 400 map with depth information of the socket housing was provided, as depicted in Figure 12. In this case, the insertion is along the robot’s *x*-axis, therefore the rotation matrix was $R=\left[\begin{array}{ccc} 0 & 0 & 1\\ 1 & 0 & 0\\ 0 & 1 & 0 \end{array}\right]$. Each unit represented 0.1 mm in robot coordinates, with the socket hole having a diameter of 4 mm. Following the protocol of previous experiments, 100 attempts were made to insert the probe into the socket, introducing random displacements of the starting point within the search area. In all attempts, the robot successfully inserted the probe into the socket in two to six attempts. The convergence and standard deviation of the search algorithm for this scenario are shown in Figure 13.

As demonstrated, the algorithm identified the target more quickly than in the previous example. This increased efficiency is attributed to the more complex environment, which provides additional information about the location during exploration.

### 4.3. Inserting a Peg into a Hole on a Conical Surface

In Section 3.1, we assumed that the 3D object consists of a finite number of horizontal faces (quasi-iso-height regions). However, the proposed algorithm can be readily extended to objects that are not composed of flat horizontal surfaces. A representative example is a cone with a hole at its apex, into which a peg must be inserted.

The key idea of the extension is to approximate inclined or curved surfaces using a series of horizontal planes. The discretization step δh is selected based on the positional repeatability of the robot. This allows us to represent arbitrary object geometries using stacked horizontal slices, to which the core algorithm from Section 3.1 and its extensions in Section 3.2 and Section 3.3 can be directly applied.

We experimentally evaluated this approach on a conical object with a base radius of 15 mm and height of 20 mm, featuring a hole with a radius of 1 mm at the apex, as shown in Figure 14. To assess the impact of height discretization resolution, we compared the convergence behavior of the algorithm for two discretization steps: δh=2 mm and δh=4 mm. The results, presented in Figure 15, demonstrate that finer discretization leads to faster convergence. Similar to previous experiments, the richer geometry afforded by finer discretization improves hypothesis elimination, thereby accelerating the search process.

These experiments demonstrate that the proposed methodology is applicable to arbitrarily inclined surfaces. However, an important open question remains: how to reliably distinguish between a contact with the hole and a contact with a region at the same height but outside the hole?

To address this, we employ a compliance-based validation procedure. The method involves executing small planar motions and observing whether the robot is constrained—an indication that the peg is within the hole. The procedure is outlined as follows:1.Define the verification plane: Construct a plane orthogonal to the estimated hole direction vector n (i.e., the insertion axis).2.Select directional vectors: Choose an arbitrary unit vector $v_{1}$ in the verification plane. Then compute a second unit vector $v_{2}$ orthogonal to $v_{1}$ within the same plane: $v_{2}=n\times v_{1}$.3.Configure robot compliance: Set the robot’s impedance controller to be compliant along both $v_{1}$ and $v_{2}$. The stiffness should be low enough to permit minor displacements without triggering safety thresholds, while still allowing detection of mechanical constraints.4.Execute test motions: Apply small, controlled displacements along $\pm v_{1}$ and $\pm v_{2}$, and monitor the actual end-effector response.5.Evaluate motion response:If no displacement is observed in either direction, the end-effector is physically constrained, indicating that the peg has entered the hole.If displacement occurs in at least one direction, the contact is not constrained, suggesting that the peg is outside the hole.6.Confirm or reject hole contact: Based on the observed response, classify the contact as a successful or unsuccessful insertion attempt.

### 4.4. Inserting the Task Board Connector into the Socket with Continuous Search

In the previous examples, we evaluated the proposed search algorithm on objects with top surfaces that were not sufficiently smooth to allow for continuous trajectory-based search. However, the proposed procedure is also applicable and efficient in cases where a continuous trajectory can be employed to systematically sweep a designated search area. To demonstrate this capability, we again utilize the Task Board, this time focusing on the insertion of the termination connector of the test probe, as shown in Figure 16.

The search procedure is initiated using a spiral search strategy, where the trajectory is continuously updated at each sampling interval t=kδt according to the following equation:(11)$$p_{t}^{r}\left(k\right)=p_{0}^{r}+\left(\delta rk\right)\left[\begin{array}{c} \sin(2\pi \gamma k)\\ \cos(2\pi \gamma k)\\ 0 \end{array}\right],$$
where $p_{0}^{r}$ represents the initial search position, δr defines the radial increment per step, and γ is the angular frequency governing the spiral motion. The parameters δr and γ must be carefully tuned to ensure that the generated trajectory sufficiently covers the search area and reliably intersects the goal region from any starting position $p_{0}^{r}$.

During the spiral search, the robot applies a controlled force in the *z*-direction while maintaining compliance along this axis. This allows it to smoothly traverse the surface and conform to any variations in height. When the probe encounters the socket opening, it slides into place, marking the successful termination of the search. Further details on controlling the robot’s stiffness and force at the tool center point can be found in [32].

To further improve the search efficiency, we integrate the spiral search with the map registration algorithm introduced in Section 3.1. First, we construct an appropriate model of the socket. Given that the plug is a cylinder with a radius of 4 mm, we account for its insertion by increasing the socket’s radius accordingly. Additionally, considering the insertion tolerance of ϵ=2 mm, the total radius of the socket hole is adjusted to accommodate this clearance. For simplification, we model the plug as a point mass while ensuring that its physical constraints within the socket are maintained (see Figure 16, right). The map registration algorithm runs concurrently with the spiral search, refining the position estimate dynamically. Specifically, an update is triggered whenever the distance between two consecutive points exceeds a predefined threshold:(12)$$∥p_{k}^{r}-p_{k-1}^{r}∥>\delta_{min}.$$

The algorithm continuously tracks the area of the current search region $S_{s}$, which contains the initial search point $p_{0}^{r}$. If the area of $S_{s}$ shrinks below the area of the goal region (i.e., the required region for successful insertion), the next command position is determined using Equation (Equation 2). In this experiment, we applied a probabilistic map registration algorithm. This adaptive refinement significantly enhances search efficiency.

The advantages of the combined search approach are illustrated in Figure 17, which compares the performance of the combined algorithm with the standard spiral search for two different initial positions. In both cases, the combined algorithm exhibited faster convergence to the goal region. However, the efficiency of the combined approach depends on the amount of information gained about different regions during the search. If the robot does not encounter new regions while searching, the combined algorithm performs similarly to the standard spiral search. Consequently, when the initial position is close to the goal region, there are no performance differences between the two methods. On the other hand, the spiral search requires precise tuning of the free parameters to successfully complete search. In contrast, the proposed combined search algorithm is successful even with poorly set spiral search parameters.

### 4.5. Summary of Experimental Results

Table 1 presents a consolidated overview of the algorithm’s performance across various use cases. It includes the number of trials, success rate, average number of attempts, standard deviation, average search time, and qualitative notes. This summary demonstrates the method’s robustness and efficiency under diverse conditions, including both discrete and continuous search strategies, and scenarios with or without environmental noise.

Experimental use cases are additionally described in the Appendix A (attached videos), where the exploration of the algorithm and the process of evaluating the starting point can be observed. The MATLAB source code of the registration algorithms in the simulated environment and videos are available via https://repo.ijs.si/nemec/3d-object-pose-detection-using-active-touch-sensing, accessed on 2 April 2025.

## 5. Conclusions

In this study, we introduced a novel algorithm for locating openings in peg-in-hole assembly tasks using sparse tactile feedback. Building upon principles from NBT techniques, particle filters, iterative tactile probing, and active hypothesis testing, the method leverages prior geometric knowledge of the target object to enable efficient search in environments with limited sensory data. Our experimental results demonstrate two key insights: (1) the algorithm achieves rapid convergence, particularly in complex environments, and (2) environmental complexity paradoxically enhances search efficiency by providing richer tactile cues that accelerate hypothesis elimination. This phenomenon arises because intricate geometries introduce distinct contact signatures, enabling the algorithm to discard incorrect hypotheses more quickly than in simpler, less informative settings.

The core algorithm, originally designed for 2D localization, was extended to 3D through innovative hypothesis confirmation and rejection protocols. By decoupling positional and orientational search dimensions, our 3D implementation avoids the curse of dimensionality, achieving computational efficiency comparable to the 2D case while improving robustness. Furthermore, we developed a probabilistic framework to address real-world challenges such as sensor noise and inaccuracies in prior maps, thereby enhancing reliability under practical conditions. The probabilistic algorithm has demonstrated significantly greater resilience to environmental noise compared to the deterministic variant. Aside from its slightly increased computational cost, it introduces no drawbacks in terms of convergence or success rate. We also demonstrated the algorithm’s compatibility with continuous-time search strategies, enabling hybrid approaches that combine the precision of tactile probing with the efficiency of motion-planning techniques.

Although the proposed algorithm is specifically designed for objects composed of multiple parallel surfaces at distinct heights, it can be readily adapted to handle objects with arbitrary geometry. This generalization requires only a simple discretization of the object model along height intervals δh, while the core algorithm remains unchanged.

While the current implementation focuses on positional localization, the architecture naturally extends to full 6-DOF pose estimation through systematic expansion of the hypothesis space. Initial investigations suggest that detecting object orientation is more challenging than position estimation and generally requires a greater number of search samples. For this reason, our future work will pursue a hierarchical approach, where the position of the hole is estimated first, followed by a finer orientation search required for full insertion of the peg.

Additional research directions include the integration of force-torque sensing for contact-rich environments and validation in industrial assembly scenarios involving variable friction, compliance, or material properties.

Although the experiments in this study relied solely on tactile sensing, the proposed method is also applicable to alternative modalities, such as laser distance sensors for non-contact probing. In such cases, the algorithm can replace exhaustive scanning procedures with a structured and efficient search strategy. Exploring this extension is part of our planned future work.

The algorithm’s ability to turn environmental complexity into a computational advantage suggests broad applicability beyond peg-in-hole scenarios. Potential applications include microsurgical robotics, where tactile feedback is essential, and space-constrained maintenance tasks in aerospace systems. By bridging geometric priors, probabilistic reasoning, and active exploration, this work contributes a principled and generalizable framework for contact-based robotic perception and manipulation in sensor-limited environments.

## Figures and Tables

**Figure 1 sensors-25-04567-f001:**
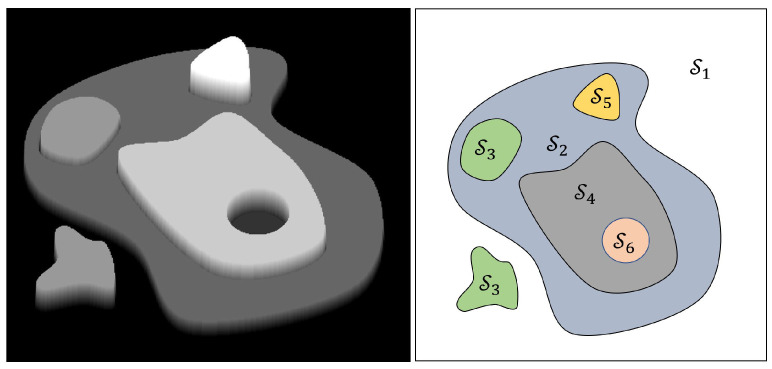
**Left**: 3D representation of the object’s surface. **Right**: A 2D map with color-coded regions $S_{i}$ based on their height. Note that some regions may be disjointed (e.g., $S_{3}$).

**Figure 2 sensors-25-04567-f002:**
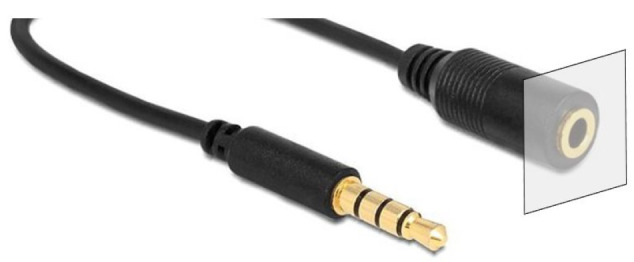
Audio plug and socket used in the example. The shaded square determines the search area for insertion of the pin into the socket and corresponds to the black area in Figure 3.

**Figure 3 sensors-25-04567-f003:**
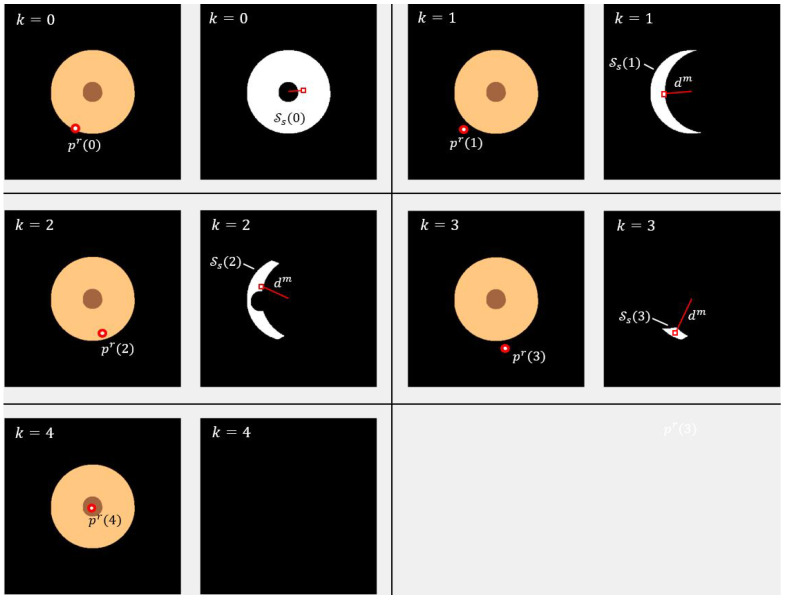
Example of the search process for the audio plug socket with progressive refinement of the search region over five steps (k=0,1,2,…). Each step includes two views: the left shows the object map with the robot’s contact point (red circle), which is unknown to the algorithm and displayed only for illustration; the right shows the current search region $S_{s}\left(k\right)$ in white, the estimated point $p_{m}^{e}$ as a red square, and the direction vector $d_{m}$. The map includes three regions: the dark brown socket hole (target), the light brown enclosure, and a black area where the robot misses the socket. At k=0, the robot touches the object, and the algorithm identifies the touched region $S_{s}\left(0\right)$. It selects $p_{m}^{e}$ near the centroid of $S_{s}\left(0\right)$ and computes $d_{m}$ toward the goal point $p_{m}^{g}$, located at the center. This guides the next move to $p_{r}\left(1\right)$.The touched region is updated using Equation (Equation 3), shrinking the search area to $S_{s}\left(1\right)$. The process repeats, with the algorithm refining $p_{m}^{e}$ and $d_{m}$ at each step, until the robot reaches the goal region $S_{g}$ at k=4, where the search area converges to zero.

**Figure 4 sensors-25-04567-f004:**
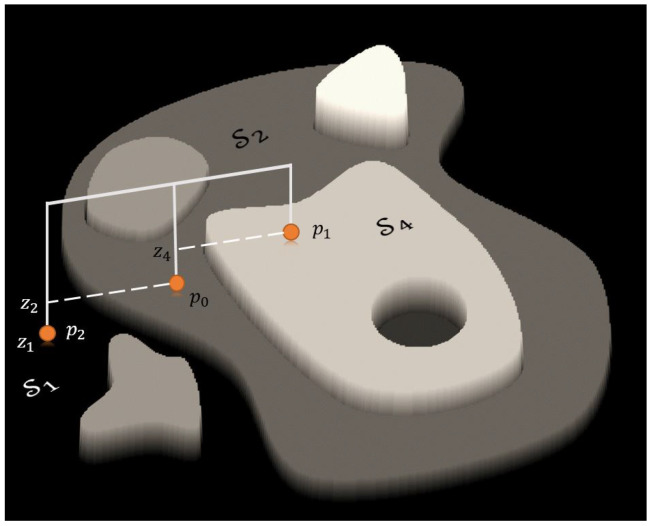
A set of contact points uniquely determines the identity of each plane. In this example, $p_{0}$ was the first touch, $p_{1}$ the second, and $p_{2}$ the third. This sequence, along with the detected height differences that identify the planes, is consistent only if $p_{0}$ belongs to region $S_{2}$.

**Figure 5 sensors-25-04567-f005:**
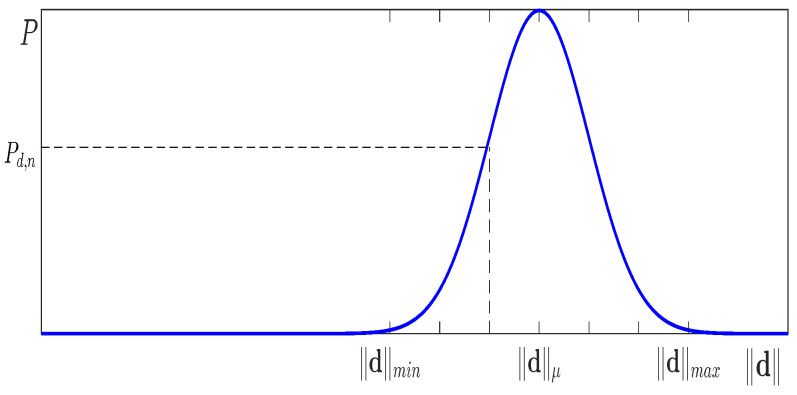
A distance ∥d∥ is modeled to be normally distributed. We sample the probability for each discrete distance $∥d_{n}∥$ in the interval from $∥d_{min}∥$ to $∥d_{max}∥$.

**Figure 6 sensors-25-04567-f006:**
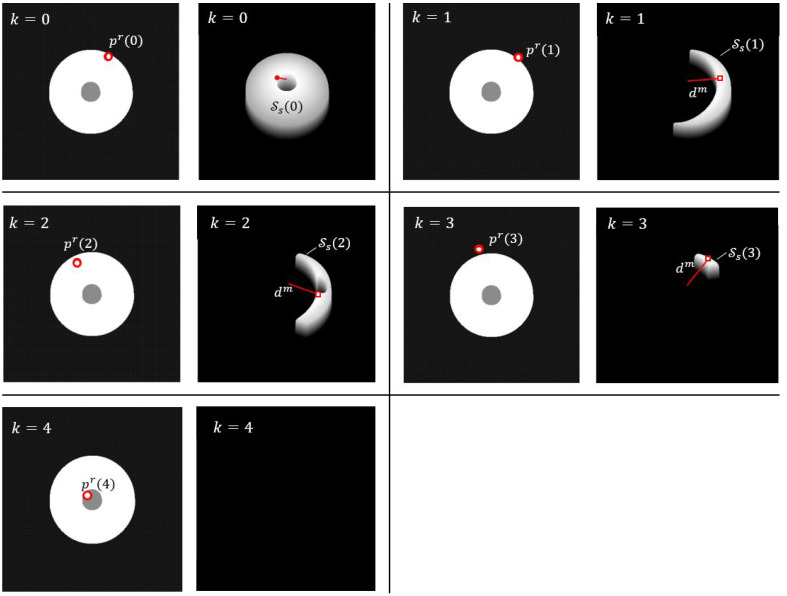
Example of probabilistic map registration for inserting an audio pin into a socket. The registration process is illustrated across sub-figures for k=0…4. In the left sub-images, the gray region represents the socket center, the white region denotes the socket body, and the black region indicates the exterior. Red dots mark contact points, which are unknown to the algorithm. In the right sub-images, the search region Ss is shown as a shaded 3D area tilted by $30^{\circ }$ around the *x*-axis, where shading intensity represents the probability estimates $P(p^{m}\in {\left(S\right)}_{s}\left(k\right)$. The red vector represents $d_{\mu }^{m}$, while the red square indicates $p_{e}^{m}$. In this probabilistic case, the search region is represented with varying probabilities of robot position, accounting for sensor noise and map inaccuracies. The transition between steps (k to k + 1) shows how the search space is adjusted dynamically, with increasing confidence in $p_{e}^{m}$.

**Figure 7 sensors-25-04567-f007:**
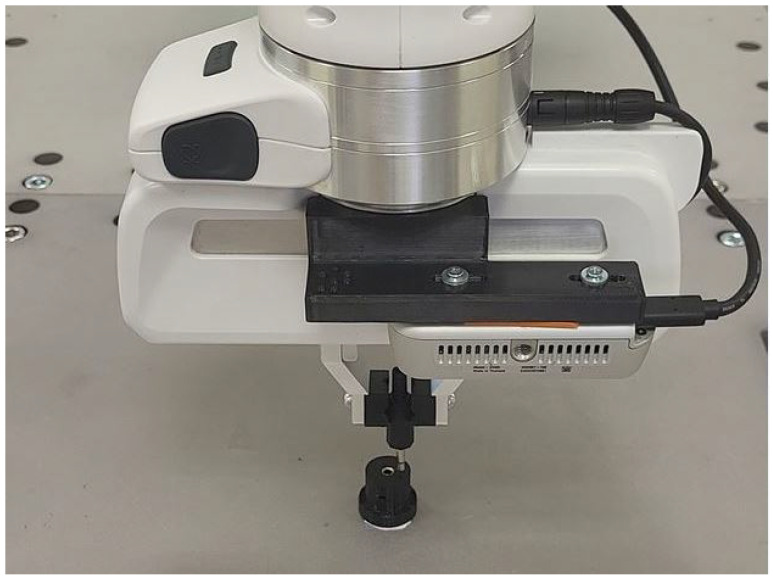
Experimental setup for testing the insertion of the audio pin into the socket.

**Figure 8 sensors-25-04567-f008:**
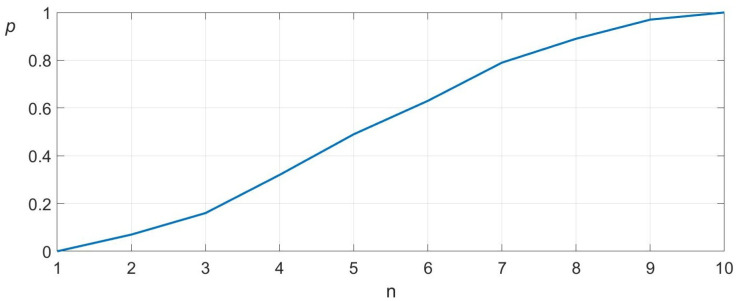
Convergence analysis of the proposed search algorithm. The *x*-axis represents the number of attempts (n), while the *y*-axis shows the probability of locating the target (*p*). The mean number of attempts is 5.83 and the standard deviation 2.04.

**Figure 9 sensors-25-04567-f009:**
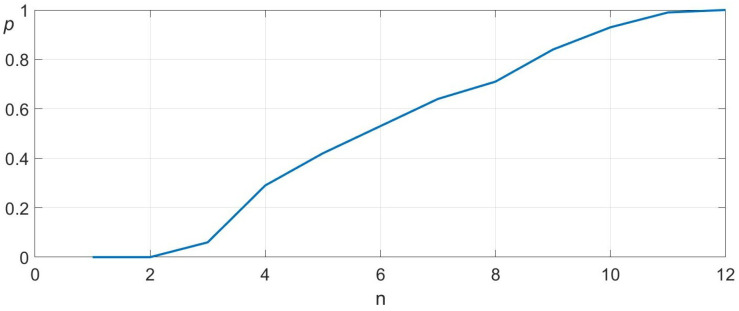
Convergence behavior of the combined search algorithm, which first determines the *z*-coordinate before localizing the *x*- and *y*-coordinates. The *x*-axis denotes the number of attempts (n) and the left *y*-axis represents the probability of hitting the target (*p*). The mean and standard deviation are 6.37 and 2.53, respectively.

**Figure 10 sensors-25-04567-f010:**
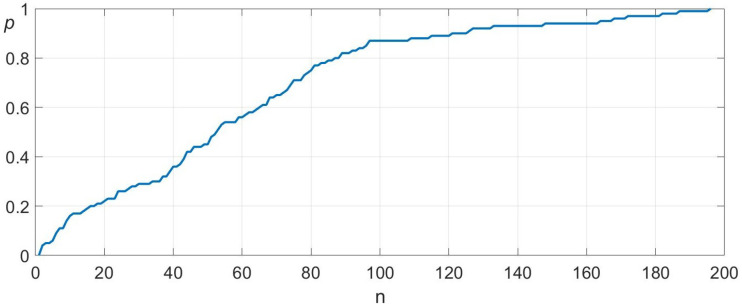
Convergence behavior of the enhanced random search. The figure shows the probability of hitting the target (*p*) related to the number of attempts (n). The mean and standard deviation are 60 and 45.5.

**Figure 11 sensors-25-04567-f011:**
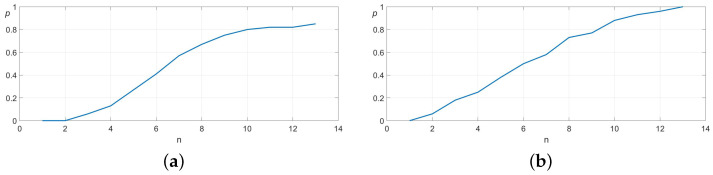
Convergence analysis of the deterministic search algorithm (**a**) and convergence analysis of the deterministic search algorithm (**b**) in noisy environments. In this case, the deterministic and probabilistic search algorithms had success rate of 85% and 100%, respectively.

**Figure 12 sensors-25-04567-f012:**
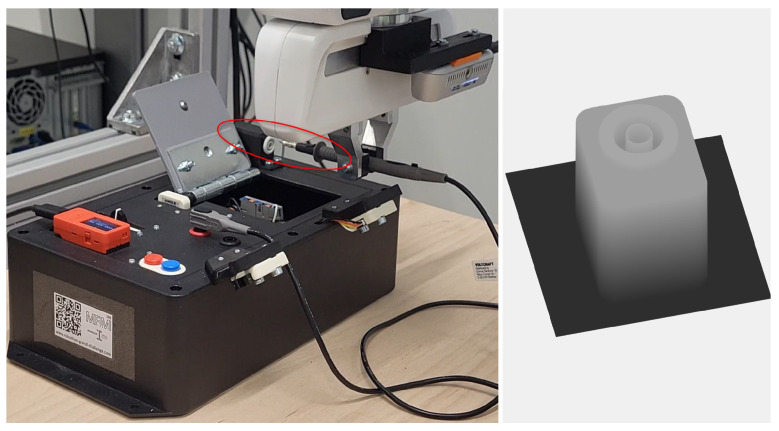
**Left**: Experimental setup for testing the stowing of the probe in the Task Board. The red oval highlights the socket and the probe. **Right**: A 3D map of the socket used for registration in the corresponding experiment. Note that the left and right images are intentionally shown from different viewpoints to emphasize the rotation R between the robot’s coordinate system and the map’s coordinate system.

**Figure 13 sensors-25-04567-f013:**
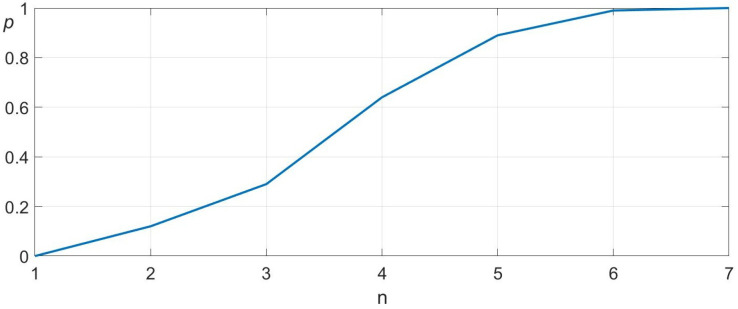
The convergence of the proposed search algorithm, showing the the probability to hit the target (*p*) vs. the number of attempts (n). Mean and the standard deviation are 4.07 and 1.18, respectively.

**Figure 14 sensors-25-04567-f014:**
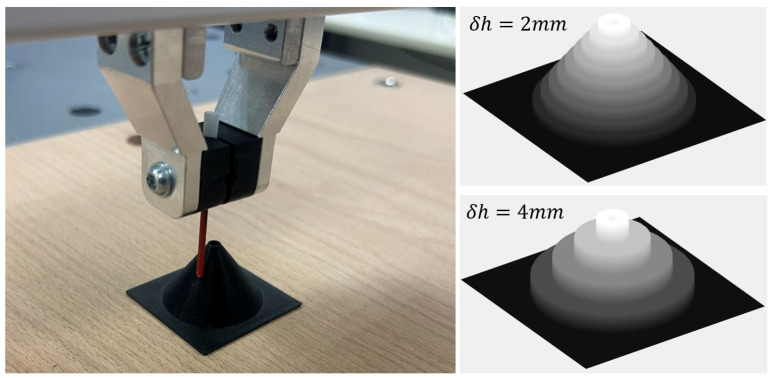
Left: Experimental setup for testing peg insertion into a hole at the apex of a cone. Right: 3D map of the cone obtained with discretization δh=2 mm and δh=4 mm.

**Figure 15 sensors-25-04567-f015:**
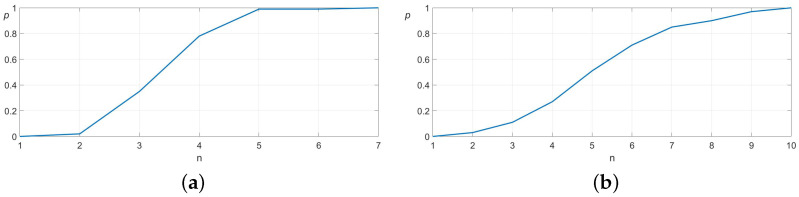
Convergence behavior of the search algorithm: the plot shows the probability of locating the target (*p*) vs. number of attempts (n). (**a**): With discretization steps of 2 mm. The mean number of attempts was 3.78 and the standard deviation was 0.84. (**b**): With discretization steps of 4 mm. The mean number of attempts was 5.65 and the standard deviation was 1.86.

**Figure 16 sensors-25-04567-f016:**
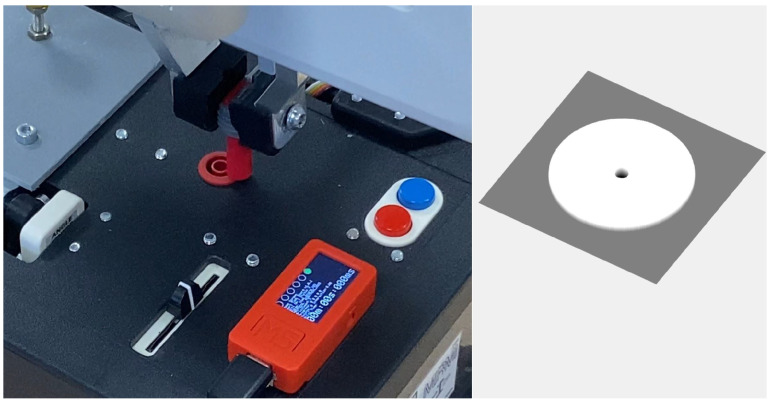
**Left**: Robot inserting the termination connector with combined spiral search and map registration algorithm. **Right**: Model of the socket, as used by the search algorithm.

**Figure 17 sensors-25-04567-f017:**
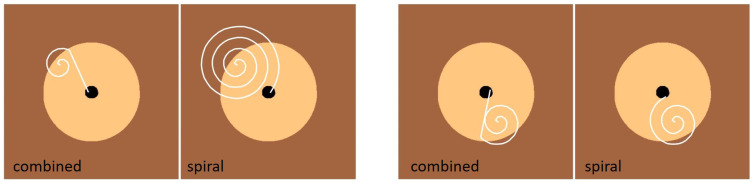
Comparison between the combined search algorithm and the spiral search algorithm for inserting the TaskBoard connector into the socket, evaluated for two different starting points. The white line represents the trajectory of the connector’s center during the search. The dark brown area indicates regions where the connector fails to engage with the socket, while the light brown area represents regions where the connector glides over the socket. The black region marks the goal.

**Table 1 sensors-25-04567-t001:** Summary of experimental results across different use cases.

Experiment	Trials	Success Rate	Avg. Attem.	Std. Dev.	Avg. Time	Notes
Audio Pin Random Search (Baseline)	100	100%	37.37	36.55	71.0 s	No prior knowledge used
Audio Pin Insertion (Deterministic)	100	100%	5.83	2.04	11.1 s	Basic algorithm with known object height
Audio Pin + Height Estimation	100	100%	6.37	2.53	12.1 s	Includes z-height search step
Audio Pin (Noisy, Deterministic)	100	85%	6.78	3.0	12.8 s	Sensitive to uncertainty; occasional failure
Audio Pin (Noisy, Probabilistic)	100	100%	6.76	2.32	12.8 s	Robust under position and map uncertainty
Task Board Probe	100	100%	4.07	1.18	8.7 s	Rich geometry improves convergence
Cone With a Hole at the Top	100	100%	3.78	0.84	7.9 s	Inclined object planes improve convergence
Task Board Connector (Combined Search)	20	100%	—	—	7.8 s	Spiral + map registration; robust to par. settings

## Data Availability

The MATLAB source code of the registration algorithms in the simulated environment and videos are available via https://repo.ijs.si/nemec/3d-object-pose-detection-using-active-touch-sensing, accessed on 2 April 2025.

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
