# Peer review of "Active Touch Sensing for Robust Hole Detection in Assembly Tasks"

_sensors, 2025, doi:10.3390/s25154567_

Round 1
Reviewer 1 Report
Comments and Suggestions for Authors
(1) For cases in 3.1 and 3.3, it is recommended to use numerical values to explain the initial position of the plug and socket, which can be omitted in 4.1 because it is a random selection.
(2) The description of experimental conditions in 4.1, such as hole size, should also be stated in the cases of 3.1 and 3.3.
(3) It is recommended to simplify the title of the picture and put the explanation of the picture content in the text.
(4) Concise the content of the summary.
Author Response
Reviewer 1
(1) For cases in 3.1 and 3.3, it is recommended to use numerical values to explain the initial position of the plug and socket, which can be omitted in 4.1 because it is a random selection.
Author Response:
We thank the reviewer for pointing this out. Accordingly, we have included the numerical displacement values in the descriptions for Sections 3.1 and 3.3. Specifically:
- In Section 3.1, the object was offset from its expected position by -6.6mm in the $x$ direction and -15.0mm in the $y$ direction.
- In Section 3.3, the displacement was 6.7mm along the $x$ axis and 15.3mm along the $y$ axis.
These values help clarify the initial conditions of each case. As noted by the reviewer, the exact initial position is not specified in Section 4.1 due to its randomized nature.
(2) The description of experimental conditions in 4.1, such as hole size, should also be stated in the cases of 3.1 and 3.3.
Author Response:
We have now added the physical dimensions of the socket and hole in the descriptions of both Sections 3.1 and 3.3 to ensure consistency. Specifically, the socket is modeled as a cylinder with a radius of 10mm and a central hole of radius 2.5mm. The search area is defined as 40 x 40 mm.
(3) It is recommended to simplify the title of the picture and put the explanation of the picture content in the text.
Author Response:
We thank the reviewer for this thoughtful suggestion regarding the presentation of Figures 3 and 6, which illustrate the search process over multiple iterations, including how the search region is reduced and how the estimated position of the first contact point is updated.
We agree that in many cases it is beneficial to move descriptive content from figure captions into the main text. However, in this specific case, we believe that keeping a detailed explanation within the figure caption is more appropriate and helpful for the reader. The captions for Figures 3 and 6 are intentionally comprehensive to allow the reader to follow the algorithm’s iterative behavior step-by-step while directly referencing the corresponding visual representation. This tight coupling between description and visual elements facilitates understanding, especially for readers who are comparing multiple sub-steps across the figure.
Relocating this content into the body text would likely result in redundancy and make the explanation less accessible, as it would force the reader to switch constantly between the figure and different parts of the text. This style—detailed multi-line captions for process figures—is also commonly used in robotics and vision literature for precisely this reason.
Nevertheless, we reviewed the captions for clarity and readability and made minor adjustments to ensure they remain concise without omitting essential information.
We hope the reviewer agrees that in this case, retaining the descriptive content in the caption best supports comprehension.
(4) Concise the content of the summary.
Author Response:
We thank the reviewer for this suggestion. We assume the comment refers to the “Summary of Experimental Results” section (Section~4.4), which presents a comparative overview of our experimental findings.
In response, we revised the text accompanying the summary table to improve conciseness and reduce repetition. We now emphasize key findings and eliminate redundant commentary. The table itself has been retained, as it provides an important high-level overview of performance across all evaluated scenarios, which we believe is valuable to the reader.
We hope the revised summary better meets the reviewer’s expectations for clarity and brevity.

Reviewer 2 Report
Comments and Suggestions for Authors
The paper presents an innovative algorithm designed to identify openings in peg-in-hole assembly tasks by utilizing sparse tactile information. Experimental findings highlight two important observations: (1) the proposed algorithm demonstrates fast convergence, especially within complex environments, and (2) greater environmental complexity unexpectedly improves search efficiency by offering more informative tactile feedback, which helps eliminate false hypotheses more quickly. The manuscript aligns well with the journal’s scope but currently requires substantial revision to enhance its overall quality and readability. The following suggestions have been provided for improvement.
1) The mathematical and algorithmic sections of the paper, particularly Sections 3.1 through 3.3, are dense and may be challenging for readers who are not deeply familiar with tactile sensing, coordinate transformations, or robotic control systems. While the technical rigor is commendable, the presentation could benefit from additional scaffolding to aid comprehension. A simplified flowchart or a high-level pseudocode summary of the entire algorithmic pipeline would help readers grasp the overall structure before diving into the details.
2) The proposed method assumes the availability of a precise 3D map of the object and, in many cases, a known orientation of the object relative to the robot. These assumptions may not hold in many real-world scenarios, especially in unstructured or dynamic environments. The paper would benefit from a more explicit discussion of these limitations and how they affect the generalizability of the approach. It would also be valuable to explore how the algorithm might be adapted or extended to handle partial or inaccurate maps, perhaps through integration with SLAM techniques or online map refinement strategies.
3) The probabilistic extension of the algorithm introduces a Gaussian distribution to model displacement uncertainty, with parameters such as σ_d chosen to cover 20–30% of the nominal displacement length. However, the rationale behind this specific choice is not empirically justified within the paper. To strengthen the argument, the authors should include a sensitivity analysis or ablation study that explores how different values of σ_d affect the algorithm’s performance. This would provide readers with a clearer understanding of the trade-offs involved and offer guidance for tuning the algorithm in different settings.
4) The experimental validation focuses on a limited set of geometries, such as audio jacks and Task Board components. While these are practical and relevant examples, they may not fully capture the diversity of shapes and configurations encountered in real-world assembly tasks. The paper would be strengthened by including experiments on more complex or irregular geometries to test the scalability and generalization of the algorithm. Additionally, a discussion on how the method scales with increasing object complexity or map resolution would be valuable.
Author Response
Reviewer 2
The paper presents an innovative algorithm designed to identify openings in peg-in-hole assembly tasks by utilizing sparse tactile information. Experimental findings highlight two important observations: (1) the proposed algorithm demonstrates fast convergence, especially within complex environments, and (2) greater environmental complexity unexpectedly improves search efficiency by offering more informative tactile feedback, which helps eliminate false hypotheses more quickly. The manuscript aligns well with the journal’s scope but currently requires substantial revision to enhance its overall quality and readability.
Author Response:
We appreciate the reviewer’s positive assessment of our work and their constructive feedback. Below, we address each comment in detail.
The following suggestions have been provided for improvement.
1) The mathematical and algorithmic sections of the paper, particularly Sections 3.1 through 3.3, are dense and may be challenging for readers who are not deeply familiar with tactile sensing, coordinate transformations, or robotic control systems. While the technical rigor is commendable, the presentation could benefit from additional scaffolding to aid comprehension. A simplified flowchart or a high-level pseudocode summary of the entire algorithmic pipeline would help readers grasp the overall structure before diving into the details.
Author Response:
We appreciate the reviewer’s thoughtful comment highlighting the need for additional scaffolding to improve clarity in the algorithmic sections. While we already provide a high-level pseudocode summary (Algorithm 1) that captures the core logic of our approach, we agree that an introductory conceptual explanation would help readers better understand the flow of the algorithm before engaging with the formal details.
To address this, we have added a new explanatory paragraph immediately before the start of Section 3.1 (after line 106). This paragraph outlines the intuition behind the algorithm in plain language, helping readers unfamiliar with tactile localization or coordinate transformations to understand the key principles of the method. The added text reads:
“To help orient the reader, we briefly describe the intuition behind the proposed method before delving into the algorithmic details. The robot, equipped with a touch sensor, can detect when it contacts a surface and measure the height of the contact point. From this information, it knows which predefined surface region of the object was touched (as each has a distinct height), but not the exact x-y location within that region. By analyzing the relative distances and directions between consecutive touch points, and aligning these with a known 3D model of the object, the algorithm incrementally narrows down the possible regions where the contact could have occurred. This process continues iteratively, pruning inconsistent hypotheses and refining the estimated position. Once the robot has localized one of the contact points with sufficient confidence, it can infer the relative position of the goal (e.g., the center of a hole) and successfully complete the insertion.”
Additionally, we have added a clarifying note immediately before Algorithm 1 to explicitly connect the conceptual explanation to the pseudocode:
“The steps in Algorithm 1 correspond to the iterative narrowing process described earlier. At each iteration, the robot uses its latest touch input to update its hypothesis about the object's position by eliminating physically inconsistent regions based on the known geometry of the object.”
We believe these additions address the reviewer’s request for improved accessibility and structural clarity without introducing possibly redundant visual elements such as a flowchart.
2) The proposed method assumes the availability of a precise 3D map of the object and, in many cases, a known orientation of the object relative to the robot. These assumptions may not hold in many real-world scenarios, especially in unstructured or dynamic environments. The paper would benefit from a more explicit discussion of these limitations and how they affect the generalizability of the approach. It would also be valuable to explore how the algorithm might be adapted or extended to handle partial or inaccurate maps, perhaps through integration with SLAM techniques or online map refinement strategies.
Author Response:
We thank the reviewer for this important comment regarding the assumptions underlying our method. We agree that the availability of a precise 3D object model is a key prerequisite. In many industrial applications, however, this requirement is practical and realistic. Object geometry can typically be obtained from CAD files, which are commonly available in structured environments. If not, the geometry can be captured using 3D scanning systems (e.g., structured light or depth cameras), many of which are commercially available and widely used.
The focus of our work is precisely on scenarios where such a geometric model is available, but visual sensing (e.g., cameras) is ineffective due to occlusions, poor lighting, or mounting constraints—e.g., when the camera is on the robot arm but cannot see the object from a useful angle. In these cases, our algorithm provides a robust alternative to full scanning by using tactile sensing to rapidly localize the relevant features with far fewer measurements.
We would also like to note that our approach is not inherently limited to binary tactile sensors. It could be adapted to use other contact-based modalities, such as line-based laser profilers, which can provide sparse geometric measurements along a path. In such cases, our algorithm could guide the laser scan dynamically, reducing the need for full-scene reconstruction and enabling faster localization through targeted probing.
We have added a brief discussion of these practical considerations and the potential for sensor generalization to the revised manuscript in the concluding section.
3) The probabilistic extension of the algorithm introduces a Gaussian distribution to model displacement uncertainty, with parameters such as σ_d chosen to cover 20–30% of the nominal displacement length. However, the rationale behind this specific choice is not empirically justified within the paper. To strengthen the argument, the authors should include a sensitivity analysis or ablation study that explores how different values of σ_d affect the algorithm’s performance. This would provide readers with a clearer understanding of the trade-offs involved and offer guidance for tuning the algorithm in different settings.
Author Response:
We appreciate the reviewer’s insightful observation regarding the choice of σ_d in our probabilistic model. The primary motivation for introducing the probabilistic framework was to improve robustness against uncertainties arising from sensor noise, map inaccuracies, and robot positioning errors—factors that can cause the deterministic algorithm to eliminate feasible regions as physically impossible due to small deviations.
While the deterministic algorithm performs well in most cases, it may fail in the presence of such deviations. In contrast, the probabilistic extension assigns lower (but non-zero) probabilities to uncertain or ambiguous situations, enabling the algorithm to maintain plausible hypotheses and continue converging even under significant uncertainty.
The choice of σ_d as 20–30% of the nominal displacement is based on empirical testing and was found to balance convergence efficiency and robustness. Although we agree that a detailed sensitivity analysis would be beneficial, we have already demonstrated the practical advantage of the probabilistic method in Section 4.1, where it outperforms the deterministic algorithm under noisy conditions. We have clarified this rationale and trade-off in the revised manuscript.
To reflect this, we added updated the introduction of Section 3.3:
"In real-world applications, robotic systems are often subject to various sources of uncertainty, including sensor noise, imperfect object maps, and calibration errors. While the deterministic version of the algorithm can tolerate moderate noise, it may fail when such deviations lead to the elimination of valid regions due to small inconsistencies. To address this, we introduce a probabilistic extension that models displacement as a distribution rather than a fixed value. Instead of rejecting inconsistent hypotheses outright, the probabilistic method assigns lower probabilities to less likely contact interpretations, allowing the algorithm to remain robust even when observations are noisy or partially inconsistent. This approach improves resilience without requiring major structural changes to the algorithm."
4) The experimental validation focuses on a limited set of geometries, such as audio jacks and Task Board components. While these are practical and relevant examples, they may not fully capture the diversity of shapes and configurations encountered in real-world assembly tasks. The paper would be strengthened by including experiments on more complex or irregular geometries to test the scalability and generalization of the algorithm. Additionally, a discussion on how the method scales with increasing object complexity or map resolution would be valuable.
Author Response:
We thank the reviewer for this important and constructive observation. To address this concern, we added a new experimental subsection (Section 4.4), where we demonstrate the applicability of our method to a more complex, non-planar geometry—a conical surface with a hole at the apex. This scenario illustrates that the proposed algorithm can handle continuous, sloped, or curved surfaces, not just objects composed of discrete parallel planes. The experiment also shows how the choice of discretization resolution () affects convergence, thus partially addressing the question of scalability with respect to map resolution.
We acknowledge that the current implementation is still limited to circular pegs and holes. Handling more complex hole geometries—such as rectangular or polygonal openings—typically requires estimation of both position and orientation. Since our current work focuses solely on positional localization, orientation estimation is not yet included. However, this is precisely the direction of our ongoing research. As outlined in the conclusion, our future approach will first determine the position of the hole (potentially by tilting the peg to maximize contact information), and only afterward refine the orientation to enable successful insertion. This staged strategy is intended to preserve the efficiency of the current method while extending it to full 6-DOF pose estimation.
We believe that the addition of Section 4.4, together with the discussion in the conclusion, addresses the reviewer’s concerns and clarifies the method’s path toward generalization and increased geometric complexity.

Reviewer 3 Report
Comments and Suggestions for Authors
In this paper, the authors propose an active touch sensing algorithm designed for robust hole localization in 3D objects. The work is interesting. I recommend this manuscript for publication after addressing the following comments.
1. Figures 8, 9, 10, 11, and 13 lack labels for the x-axis and y-axis. These axes should be clearly labeled within the figures. Furthermore, the placement of labels (a) and (b) in Figure 11 is potentially misleading. These labels should be repositioned to the upper-left corner of the respective subfigures.
2. The size of Figure 13 is inconsistent with the sizes of other figures in the manuscript. The author should standardize the sizes of all illustrations in the article.
3. The detection conditions in this manuscript are limited to discrete-height parallel planes. However, actual industrial components may have curved or slanted surfaces. Please address whether the detection method maintains its effectiveness when applied to such non-planar surfaces.
4. Can the detection method proposed in the manuscript distinguish between circular blind holes and circular through-holes?
5. The authors present various sensors in the “Introduction” section, including force sensors, tactile sensors, laser scanners, etc. The relevant references are suggested to be cited, such as (1) Ultrastrong and heat-resistant self-powered multifunction ionic sensor based on asymmetric meta-aramid ionogels Chemical Engineering Journal, 2025, 519, 165332 (https://doi.org/10.1016/j.cej.2025.165332).
Author Response
Reviewer 3
In this paper, the authors propose an active touch sensing algorithm designed for robust hole localization in 3D objects. The work is interesting. I recommend this manuscript for publication after addressing the following comments.
Author Response:
We appreciate the reviewer’s positive assessment of our work and their constructive feedback. Below, we address each comment in detail.
- Figures 8, 9, 10, 11, and 13 lack labels for the x-axis and y-axis. These axes should be clearly labelled within the figures. Furthermore, the placement of labels (a) and (b) in Figure 11 is potentially misleading. These labels should be repositioned to the upper-left corner of the respective subfigures.
Author Response:
We have added the missing x-axis and y-axis labels to Figures 8, 9, 10, 11, and 13. Additionally, we have repositioned the (a) and (b) labels in Figure 11 to the upper-left corner of each subfigure to ensure clarity.
- The size of Figure 13 is inconsistent with the sizes of other figures in the manuscript. The author should standardize the sizes of all illustrations in the article.
Author Response:
We have standardized the sizes of all convergence plots throughout the manuscript for consistency. The only exceptions are Figures 11 and 15, where side-by-side comparison of two methods necessitates a wider layout.
- The detection conditions in this manuscript are limited to discrete-height parallel planes. However, actual industrial components may have curved or slanted surfaces. Please address whether the detection method maintains its effectiveness when applied to such non-planar surfaces.
Author Response:
We thank the reviewer for highlighting this important point. In the revised manuscript, we have added a new subsection (Section 4.4) titled Inserting a Peg into a Hole on a Conical Surface, in which we explicitly address the generalization of our method to curved and slanted surfaces.
In this section, we demonstrate that the proposed algorithm can be readily extended to arbitrary geometries by approximating curved or inclined surfaces with a set of horizontal planes. The height discretization step $\delta h$ can be chosen based on the positional repeatability of the robot. We validated this approach experimentally on a conical surface with a hole at its apex, using two different discretization resolutions. The results show that the method not only remains effective on non-planar surfaces but that increased geometric complexity—when adequately represented—actually improves convergence. This is consistent with earlier findings that richer geometric structure provides more informative tactile cues for hypothesis pruning.
- Can the detection method proposed in the manuscript distinguish between circular blind holes and circular through-holes?
Author Response:
We thank the reviewer for raising this important practical consideration. The current implementation uses a physical probe for contact detection, so the ability to distinguish blind from through-holes is constrained by the probe’s length. If the hole is deeper than the probe’s sensing limit, differentiation becomes infeasible with this hardware.
However, this limitation can be addressed by integrating non-contact sensors (e.g., laser distance sensors), which would extend sensing depth and allow detection of through-holes. Although not implemented in this study, we have mentioned this as a direction for future work.
More broadly, when a hole and a flat surface exist at the same height—especially on inclined surfaces—the distinction becomes more challenging. To tackle this, we introduced a compliance-based validation procedure at the end of Section~4.4. This method tests for physical constraint in directions orthogonal to the insertion axis. If the end-effector is restricted in both directions, the contact is identified as a successful insertion. This additional check enhances robustness and supports distinction between valid hole insertions and surface contacts.
- The authors present various sensors in the “Introduction” section, including force sensors, tactile sensors, laser scanners, etc. The relevant references are suggested to be cited, such as (1) Ultrastrong and heat-resistant self-powered multifunction ionic sensor based on asymmetric meta-aramid ionogels Chemical Engineering Journal, 2025, 519, 165332 (https://doi.org/10.1016/j.cej.2025.165332).
Author Response:
We have incorporated the suggested reference into the “Introduction” section, along with three additional relevant works to strengthen the background on tactile and force sensing technologies.

Round 2
Reviewer 2 Report
Comments and Suggestions for Authors
The authors have addressed the concerns of the reviewer satisfactorily. Hence, recommend the article for publication.